# Multianalytical Study of the Blue Pigments Usage in Serbian Iconography at the Beginning of the 18th-Century

Maja Gajić-Kvaščev [1,*], Olivera Klisurić [2], Velibor Andrić [1], Stefano Ridolfi [3], Una Galečić [4] and Daniela Korolija Crkvenjakov [5]

1   Department of Chemical Dynamics and Permanent Education, National Institute of the Republic of Serbia, Vinča Institute of Nuclear Sciences, University of Belgrade, 11001 Belgrade, Serbia; velan@vinca.rs
2   Department of Physics, Faculty of Sciences, University of Novi Sad, 21000 Novi Sad, Serbia; olivera.klisuric@df.uns.ac.rs
3   Ars Mensurae, 00188 Roma, Italy; arsmensurae@gmail.com
4   The Provincial Institute for the Protection of Cultural Monuments, 21131 Petrovaradin, Serbia
5   Academy of Arts, University of Novi Sad, 21000 Novi Sad, Serbia; daniela.korolija@gmail.com
*   Correspondence: gajicm@vinca.rs

**Abstract:** Traditional Serbian religious art originated in Byzantine culture and conserved Byzantine elements until modern times. However, since the end of the 17th century, many changes in traditional icon painting have been introduced. Previous studies focused on the changes in iconography and style, but very little attention was paid to the changes in painting materials and techniques. This research focuses on the blue pigments on icons from the first half of the 18th century. Eight icons with blue areas of a different hue were selected for the study. Due to its rarity in nature, price, and iconographical importance, the blue pigment was particularly praised by painters. Therefore, the choice of the blue pigment can be related to historical information to trace influences and the development of the painter's practice. Imaging techniques, several portable, non-destructive analytical techniques, such as XRF and FTIR, followed by optical microscopy and SEM-EDX analysis of the samples were used to characterize blue pigments and the painting technique. An analysis showed that icon painters at the beginning of the 18th century used pigments such as azurite, an indigo-organic colourant of plant origin, and Prussian blue. Contrary to the traditional belief that natural ultramarine blue was used, it was not confirmed in any studied examples.

**Keywords:** blue pigment; XRF; FTIR; SEM-EDX; icons

## 1. Introduction

Icons are the most diffused objects of devotion in Serbian religious art. Stylistically and technically rooted in the Byzantine medieval tradition, icons were created following strictly defined protocols and technical recipes codified by the rules of the Christian Orthodox church and pre-set in the painting manuals like Hermeneia of the Art of Painting, compiled by the hieromonk and icon-painter Dionysius of Fourna c.1730 [1,2].

Traditional Serbian icon painters used typical medieval materials and techniques: wooden supports, chalk and gypsum for the preparation layer, egg tempera paint, and gold and silver leaves. They learned painting either in the monasteries, which at that time, were the only official places to learn the art of icon painting, or they were trained and worked inside small workshops. The information about their lives and work is scarce, so attribution to a certain workshop is often based on stylistic characteristics [3,4].

The icon painting did not substantially change in the post-Byzantine period. The break with tradition and the acceptance of the Baroque influences follows the migrations of Serbs at the end of the 17th century, from the central Balkans to the Habsburg Empire, with dominated Baroque artistic production. Two distinctive styles, the Byzantine and the Baroque, coalesced in the works of painters, creating a unique hybrid style in Serbian icon

painting in the last decade of the seventeenth and the first half of the eighteenth century, the so-called *zographic* style. Therefore, although icons were considered religious art, rigidly stereotyped by the heavy hand of church and tradition, various historical and territorial influences made the icon often chosen as the means of launching new painting materials and ideas in art. This fact triggered the current scientific investigation of the materials of the *zographic*-styled icons, covering the period from 1700 to 1750, from the collection of the Gallery of Matica Srpska in Novi Sad (Serbia), the museum that holds one of the richest collections of Serbian 18th-century religious art [5].

The present study puts focus on blue pigment because it was one of the most expensive, and for painters working in a modest economic environment, it is reasonable to expect to search for cheaper alternatives. Also, the 18th-century painters, alongside traditional blue pigments of natural origin, had the possibility to use Prussian blue as the first synthetic blue. Moreover, the choice of blue pigments is a good indicator of the painter's choices of materials according to price and availability. In many examples, blue is completely avoided. Instead, a greyish hue could be noticed, satisfying the iconographical needs without using blue pigments.

Different analytical approaches were successfully employed in the examination of the icons [6–12]. The examinations of the Serbian icons in the context of the used materials and painting techniques are rather rare [13–17], and mostly unpublished.

The aim of this study is to use non-destructive analytical techniques to characterize the materials used on the blue parts of the icons. The multi-analytical approach, including imagining techniques: ultraviolet reflected photography (UVR), ultraviolet luminescence photography (UVL), infrared reflectography images (IRR), infrared false Ccolour (IRFC) and non-destructive energy dispersive X-ray fluorescence (EDXRF). Fourier transform infrared spectrometry in reflection mode (rFTIR) was employed together with sample analysis using optical microscopy (OM) and scanning electron microscopy with energy dispersive X-ray spectroscopy (SEM-EDX). Imaging techniques are applied to determine the presence of later interventions, which ensured the examination of the original materials. Some preliminary pigment identifications were carried out using the IRFC images. The EDXRF spectrometry was used to identify inorganic pigments; these results are confirmed with additional measurements with rFTIR spectrometry techniques. The binders were identified by rFTIR analysis. There are a few cross-sections of samples of blue pigments available as additional information about the stratigraphy of the layers, for a final discussion about the painting technique. The SEM-EDX offered detailed analysis through paint layers, enabling more reliable identification of the used materials. Gathered analytical data led to the identification of the blue pigments in the selected Serbian icons from the first half of the 18th century.

## 2. Materials and Methods

### 2.1. Icons

This study was conducted on eight *zograf* icons from the collection of the Gallery of Matica Srpska, Novi Sad, Serbia. Selected icons cover the period from 1700 to 1750 and belong to the traditional icon painting. The painting technique is egg tempera on a wooden panel. The icons were selected among the most representative group of traditional 18th-century icons, chosen by the museum curators. The second criteria were the presence of different hues of blue colour in the paint. Four icons are attributed to known authors, and the authors of the other four icons are anonymous. The relevant data about the icons are presented in Table 1.

**Table 1.** The historical data of the *zograf* icons selected for the study.

| Museum Label GMSU | Icon Attribution | Dimensions [cm] | Date |
|---|---|---|---|
|  6334 | *Saint Theodore Tyron and Saint George* Anonymous painter | 95.5 × 65 | ca. 1700. |
|  6112 | *Saint Demetrius* Hristofor Žefarović | 89 × 72 | ca. 1735 |
|  6215 | *Jesus Christ* Anonymous painter | 100 × 66 | ca. 1750 |
|  6472 | *The Virgin and Child* Anonymous painter | 95.5 × 80 | 1742 |
|  6506 | *The Annunciation with the Prophets David and Solomon* Anonymous painter | 148 × 76 | First quarter of the 18th century |
|  6432 | *Jesus Christ* Georgije Stojanović | 100 × 75 | 1737 |
|  1451 | *St. John the Baptist with Scenes from His Life* Stanoje Popović | 100 × 74 | ca. 1744 |
|  1586 | *Jesus Christ with the Apostles in Medallions* Stanoje Popović | 124.5 × 82.5 | 1743 |

## 2.2. Imaging Techniques

Imaging techniques have been used to identify homogeneities/heterogeneities in the icons. Moreover, these observations helped choose the most interesting areas for spectroscopic analyses and the most suitable areas for sampling.

Prior to instrumental analysis, icons were observed under UV light (wood lamps) to detect non-original paint, thus ensuring the examination of only original pigments.

Infrared reflectography images (IRR) were captured with MICRO IR 20 reflectography camera (wavelength 400–1100 nm, 12.5 mm f1.3 lens, and VIS and IR80 filters, EIS, Rome, Italy). Furthermore, visible and IRR images were combined using Adobe Photoshop software (version 10.0, CS3) to obtain IRFC images [18–20]. The IRFC results are reported as the specific colour of the painted layer for selected icons.

## 2.3. Non-Destructive Analysis

After the examination by imaging techniques, the icons were subjected to EDXRF and rFTIR investigation to determine as precisely as possible the composition of the materials that were used at the selected blue measuring positions. All the mentioned spectrometric analyses were performed using instruments that enabled in situ and non-destructive analytical procedures. The measuring points on the painted layer of the icons were selected to be informative for pigment detection, suitable for simultaneous EDXRF and rFTIR measurements (big enough and flat as much as possible) and sampling.

The EDXRF spectrometric technique was used to determine the elemental composition. An in-house developed milli-beam EDXRF spectrometer with a side-window X-ray tube (Oxford instruments-OXFORD Instruments, Scotts Valley, CA, USA, Rh anode, max. voltage 50 kV, max. current 1 mA, air cooled) was used. The excitation beam was collimated to an approximately 2 mm spot size on the object's surface using a layered structure pinhole collimator. The X-123, AMPTEK Inc-AMPTEK, Inc., Bedford, MA, USA. with Si-PIN detector (6 mm$^2$/500 μm, resolution 160 eV at Mn K$\alpha$ line, 12.5 μm thick Be window and 1.5″ extension) was mounted at the position of 45° to an incident beam axis. To minimize the detection of the X-rays scattered from the collimator, the detector window was placed 10 mm ahead of the tip of the collimator. Two laser pointers, ensure precise reproducible alignment of the excitation beam and visualization of the measured spot. The experimental set-up, X-ray tube voltage 40 kV, filament current 800 μA, non-filtered excitation X-rays, and measurement time 60 s were kept constant for all measurements. The spectra were acquired with ADMCA software (AMPTEK Inc., version 1, 0, 0, 16).

The rFTIR spectra at the blue positions on the icons were recorded with a Bruker Optics ALPHA-R (Bruker Optik GmbH, Ettlingen, Germany) portable infrared spectrophotometer equipped with a Globar Mid-IR source, a modified Michelson all-spatial interferometer (RockSolidTM), and a DLaTGS detector (Bruker Optik GmbH trade marks TM) at room temperature. Measurements were made using an external reflection module with optics (22°/22°), the analysed spot is about 4 mm in diameter. The analysis was performed in the range between 400 and 4000 cm$^{-1}$, by averaging 64 scans at a resolution of 4 cm$^{-1}$. The spectrum of a golden flat mirror was used as the background. The associated OPUS software package (version 7.0, Build 7.0) was used to evaluate the recorded spectra.

## 2.4. Sample Preparation and Analysis

Samples taken from icons were taken near the places with existing damage to the painting layer, so that further damage, however minimal, is hardly noticeable. Before embedding the sample in resin for microscopic investigation, macro images of the sample are documented as well. For embedding, a mixture of epoxy resin (Epoxy resin L) and hardener (Hardener EPH 161) in a 4:1.5 ratio was used (R&G Faserverbundwerkstoffe GmbH, Waldenbuch, Germany), filling the sampling mould halfway and leaving it to partially cure, then placing the sample and then overflowing the sample to fully embed it. Before the overflowing of the sample, it is manipulated with a needle to place it in an ideal position for observing the cross-section, and this manipulation is observed with a BRESCIANI

ARGO S3 microscope, Bresciani Srl, Milan, Italy. The curing time for this epoxy mixture was 24 h. Afterwards, a polishing phase, using polishing paper of different granulation (P150–P1200 range), was conducted. Polished cross-sections of samples were examined under an optical OLYMPUS BX 51M microscope (Olympus Corporation, Shinjuku, Tokyo, Japan) using dark-field normal light and UV light conditions at magnifications up to 200×. Before the SEM-EDX investigation, samples were sputter coated with gold (Bal-Tec SCD 005, BAL-TEC in now Leica Biosystems, Barrington, IL, USA) using current 30 mA during 90 s, at a distance of 50 mm for conductivity purposes. SEM images were collected on a JEOL JSM 6460 LV device-Japan Electron Optics Laboratory Co., Ltd, Tokyo, Japan, (applied voltage 20 kV at the working distance approximately 10 mm). The morphological characteristics of the pigment grains were observed by using the backscattered electrons (BSE) and the secondary electrons (SE) as well. EDX measurements were carried out on the Oxford INCA X sight detector (Oxford Instruments NanoAnalysis&Asylum Research, High Wycombe, UK).

## 3. Results and Discussion

The IRFC imagining technique and SEM-EDX analysis were performed on three representative icons (GMSU 6472, 6112 and 1451) while spectroscopic methods (EDXRF and rFTIR) were conducted on all eight icons.

The measuring spots are denoted in visible light photography (Figure 1a–c). The IRFC images of the selected icons are presented in Figure 1d–f. The results of the spectroscopic and sample analysis are presented in Figures 2 and 3, respectively describing detected components required for blue pigments' identification. Among different blue pigments, three were identified on examined icons: azurite, Prussian blue, and indigo.

The dark-purple-blue areas in the IRFC image of the icon GMSU 6472—The Virgin and Child (Figure 1d) suggest the presence of azurite. Moreover, the dark-blue nuance in the same IRFC image at the bottom area of the Virgin's dress, as well as an intense red hue, suggests the presence of some other blue pigments. An intense copper peak detected in EDXRF spectra (Figure 2a) confirmed the presence of azurite. The rFTIR lines around 460 $cm^{-1}$, 500 $cm^{-1}$, 840 $cm^{-1}$, 960 $cm^{-1}$, 1090 $cm^{-1}$, doublet 1434 and 1460 $cm^{-1}$, 1860 $cm^{-1}$, and three resolved contributions at 2500, 2553, and 2587 $cm^{-1}$, assigned bands due to stretching or bending vibrations (described in more details in Table 2 and presented in Figure 2b) once again confirm the presence of azurite [21,22]. The difference in the FTIR spectra (F1P and F6), better noticed if the experimental data treated by the Savitzky–Golay filter (small picture at Figure 2b), suggests the use of indigo pigment (characterized by doublet around 1460 and 1480 $cm^{-1}$, and lines around 1580 and 1630 $cm^{-1}$, [23]).

**Table 2.** EDXRF and FTIR data for the blue pigments that enabled its identification.

| Museum Label GMSU | EDXRF Chemical Elements | rFTIR Spectral Data [$cm^{-1}$] | Pigment Identification |
|---|---|---|---|
| 6334 | Cu * (Pb, Ca, Fe) ** | 460 [a], 500 [b], 840 [c], 960 [d], 1090 [e], doublet 1434 [f] and 1460 [f], 1860 [g] and triplet around 2500 [h], 2553 [h] and 2587 [h] | Azurite |
| 6112 | Fe, K * (Pb, Ca, Sr, Cu) ** | 2090 [i] | Prussian blue |
| 6215 | Cu * (Pb, Ca, Fe) ** | 460 [a], 500 [b], 840 [c], 960 [d], 1090 [e], doublet 1434 [f] and 1460 [f], 1860 [g] and triplet around 2500 [h], 2553 [h] and 2587 [h] | Azurite |
| 6472 | Cu * (Pb, Ca, Fe) ** | 460 [a], 500 [b], 840 [c], 960 [d], 1090 [e], doublet 1434 [f] and 1460 [f], 1860 [g] and triplet around 2500 [h], 2553 [h] and 2587 [h]; doublet around 1460 [j] and 1480 [j] and lines around 1580 [k] and 1630 [l] | Azurite and Indigo |
| 6506 | Fe * (Pb, Ca) ** | doublet around 1460 [j] and 1480 [j] and lines around 1580 [k] and 1630 [l]; 2090 [i] | Indigo and Prussian blue |

**Table 2.** *Cont.*

| Museum Label GMSU | EDXRF Chemical Elements | rFTIR Spectral Data [cm⁻¹] | Pigment Identification |
|---|---|---|---|
| 6432 | Fe, K * Cu * (Pb, Ca) ** | 2090 [i]; 460 [a], 500 [b], 840 [c], 960 [d], 1090 [e], doublet 1434 [f] and 1460 [f], 1860 [g] and triplet at 2500 [h], 2553 [h] and 2587 [h] | Prussian blue and Azurite |
| 1451 | Fe * (Pb, Ca, Sr, Cu) ** | doublet around 1460 [j] and 1480 [j] and lines around 1580 [k] and 1630 [l]; 2090 [i] | Indigo and Prussian blue |
| 1586 | Cu * (Pb, Ca, Fe) ** | 460 [a], 500 [b], 840 [c], 960 [d], 1090 [e], doublet 1434 [f] and 1460 [f], 1860 [g] and triplet around 2500 [h], 2553 [h] and 2587 [h] | Azurite |

* Chemical element which defines the pigment. ** Other detected chemical elements. Assignments of MIR bands of the pigments. Azurite: (a) $\nu$(Cu-OH), (b) $\nu$(Cu-O), (c) $\nu_2$(CO$_3^{2-}$), (d) $\delta$(OH), (e) $\nu_1$(CO$_3^{2-}$), (f) $\nu_3$(CO$_3^{2-}$), (g) $\nu_1 + \nu_4$ (CO$_3^{2-}$), (h) $\nu_1 + \nu_3$ and/or $2\nu_2 + \nu_4$ (CO$_3^{2-}$); [22]( for a–f), [21] (for g–h). Prussian blue: (i) $\nu$(C ≡ N)-cyano stretching vibration; [23]. Indigo: (j) C-H bending and C-C stretching combination, (k) CC stretching vibrations in indigo molecule, (l) C=O stretching; [23].

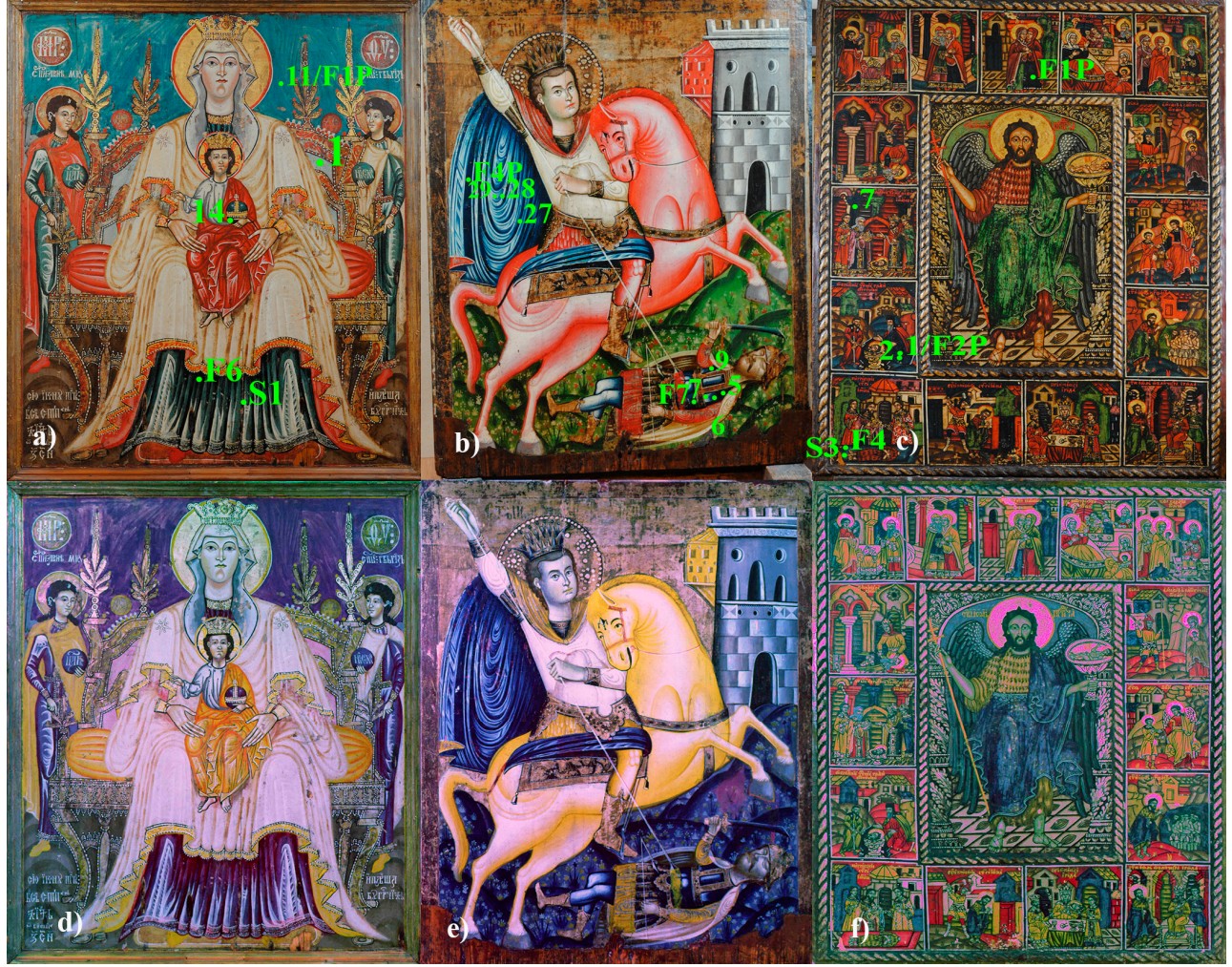

**Figure 1.** (**a**) Icon GMSU 6472—*The Virgin and Child* blue parts measuring points (EDXRF-numbered, rFTIR-F-numbered, and samping-S-numbered; (**b**) Icon GMSU 6112—*Saint Demetrius* blue parts measuring points (same notation as for the icon GMSU 6472); (**c**) Icon GMSU 1451—*St. John the Baptist with Scenes from His Life* blue parts measuring points (same notation as for the icon GMSU 6472); (**d**) The IRFC image of the icon GMSU 6472—*The Virgin and Child*; (**e**) The IRFC image of the icon GMSU 6112—*Saint Demetrius*; (**f**) The IRFC image of the icon GMSU 1451—*St. John the Baptist with Scenes from His Life*.

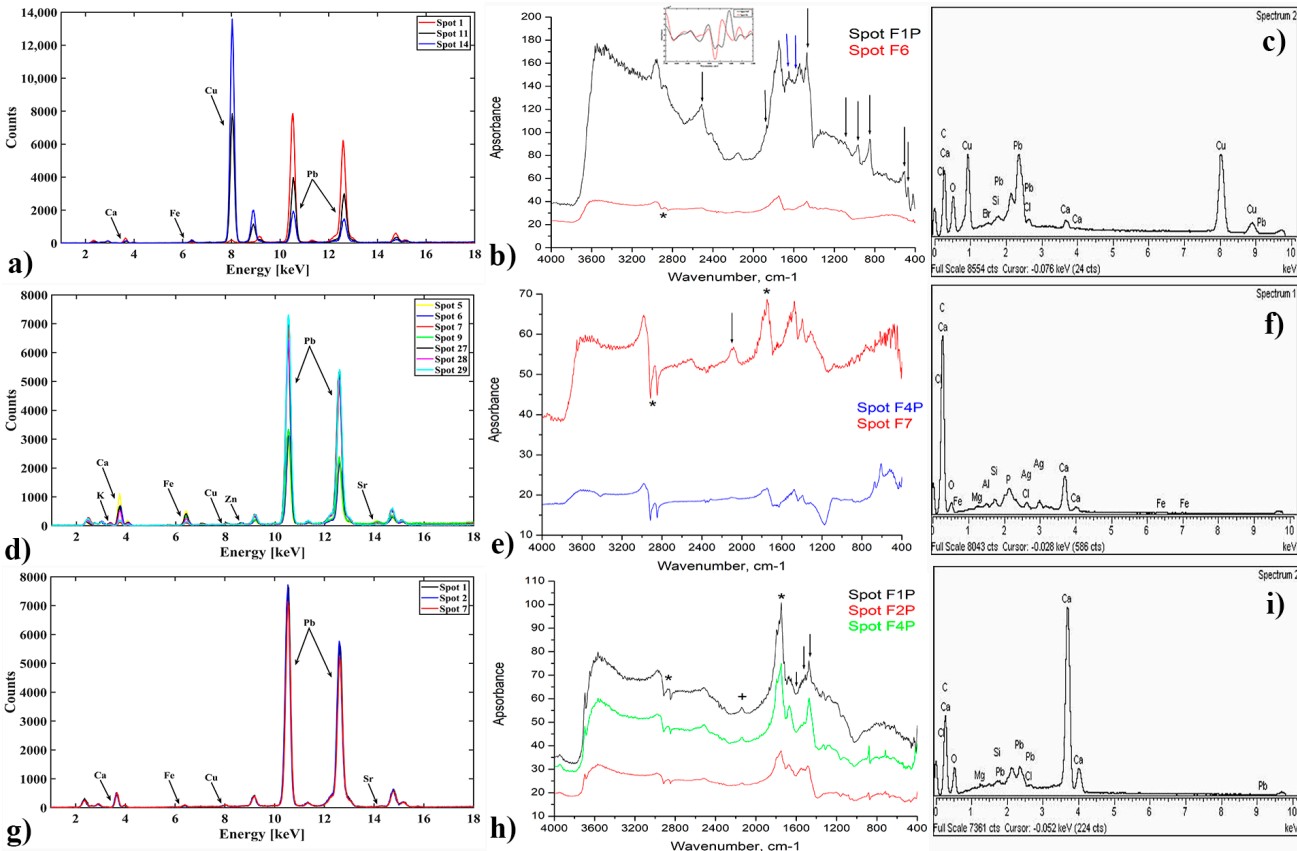

**Figure 2.** (**a**) The comparative EDXRF spectra collected at the blue parts of the icon GMSU 6472 (detailed spectra are provided in Figure S1 in the Supplementary Material); (**b**) The comparative absorbance rFTIRF spectra collected at the blue parts of the icon GMSU 6472 (black and blue arrows corresponds to the characteristic lines for azurite and indigo, respectively, while * denotes characteristic lines for egg tempera [24,25]); (**c**) SEM-EDX spectrum corresponds the blue layer in the sample S1; (**d**) The comparative EDXRF spectra collected at the blue parts of the icon GMSU 6112 (detailed spectra are provided in Figure S2 in the Supplementary Material); (**e**) The comparative absorbance rFTIRF spectra collected at the blue parts of the icon GMSU 6112 (black arrow corresponds to the characteristic line for Prussian blue, while * denotes characteristic lines for egg tempera); (**f**) SEM-EDX spectrum corresponds the blue layer in the sample S2; (**g**) The comparative EDXRF spectra collected at the blue parts of the icon GMSU 1451 (detailed spectra are provided in Figure S3 in the Supplementary Material); (**h**) The comparative absorbance rFTIRF spectra collected at the blue parts of the icon GMSU 1451 (black arrows corresponds to the characteristic lines for indigo, while + and * denotes characteristic lines for Prussian blue pigment egg tempera binder, respectively); (**i**) SEM-EDX spectrum corresponds the blue layer in the sample S3.

Sample S1 (Figure 1a) was taken from the blue Virgin's robe. The cross-section of sample S1 in the OM photography in visible light (Figure 3a) shows a clearly visible preparation layer followed by three to four coloured layers. The number of layers varies according to the position on the cross-section and depicts the brushwork during the painting process. The lowest-coloured layer A is pale blue in OM in visible light (Figure 3a) while the SEM image of the same layer (Figure 3c) shows bright white particles, indicating the mixture of azurite and lead white pigments. The middle layer B is of a more intense blue colour (Figure 3a) and has spherical shaped particles visible on the SEM image of the same layer (Figure 3c) which is typical for artificial azurite [7]. Layer B is partially overlapped with another pale blue layer (C) similar in microstructure to the grain types of layers A and B. Layer D has a very homogeneous structure and no individual pigment grains are visible (Figure 3a) indicating the use of very-fine-grained organic indigo pigment.

The confirmation of the azurite and indigo is also given by the presence of characteristic elements (copper and carbon) in the EDX spectra (Figure 2c). The mixture of the pigments can be clearly seen on the EDX spectrum collected in layer C (Figure 3a) where characteristic copper (for azurite) and carbon (for indigo) peaks can be noticed (Figure 4a). The well-established practice of using egg tempera as a binder has been confirmed by rFTIR analysis (characteristic lines at 1740 cm$^{-1}$ and doublets at 2850 and 2920 cm$^{-1}$ [24,25] are clearly visible in the FTIR spectra, Figure 2b).

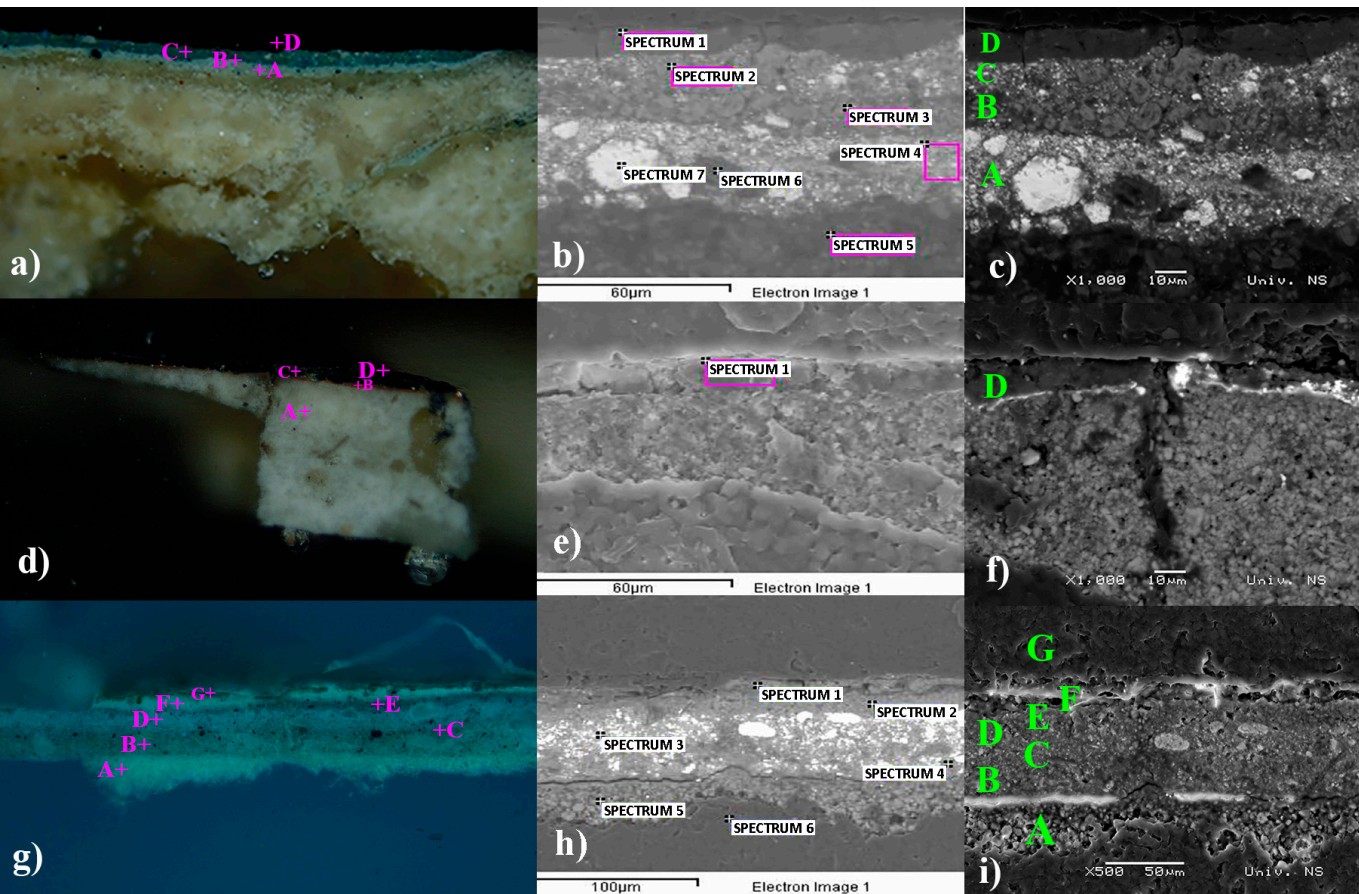

**Figure 3.** (**a**) Optical microscopy image (magnification ×100) of the sample S1; (**b**) SEM image (magnification ×1000) of the sample S1 layers; (**c**) SEM image (magnification ×1000) of the sample S1; (**d**) Optical microscopy image (magnification ×200) of the sample S2; (**e**) SEM image (magnification ×1000) of the sample S2; (**f**) SEM image (magnification ×1000) of the sample S2; (**g**) Optical microscopy image under UV light of the sample S3 (magnification ×200); (**h**) SEM image (magnification ×500) of the sample S3 layers; (**i**) SEM image (magnification ×500) of the sample S3.

Dark-blue-blackish areas in the IRFC image of the Icon GMSU 6112—*Saint Demetrius* (Figure 1e) suggest the presence of Prussian blue. The peak of iron detected in EDXRF spectra (Figure 2d) and the rFTIR line around 2090 cm$^{-1}$ assigned to cyano stretching vibration $\nu(C \equiv N)$ (Figure 2e) confirmed the presence of Prussian blue [23]. The potassium (Figure 2d) and phosphorus peaks (Figure 2f) might indicate the production technology as well provenance of the raw materials for the Prussian blue pigment production (as referred in [7], p. 787). The presence of phosphorus may also indicate using bone black pigment to intensify the blue nuance, as it was common practice in orthodox iconography at that time. The very low intensity of Cu and Zn peaks do not change with the hue of blue colour (Figures 2d,g, S2 and S3), indicating that these peaks are unrelated to the pigment composition. The Cu and Zn peaks result from the scattering due to the instrumental setup. The use of egg tempera as a binder has also been confirmed in the rFTIR analysis.

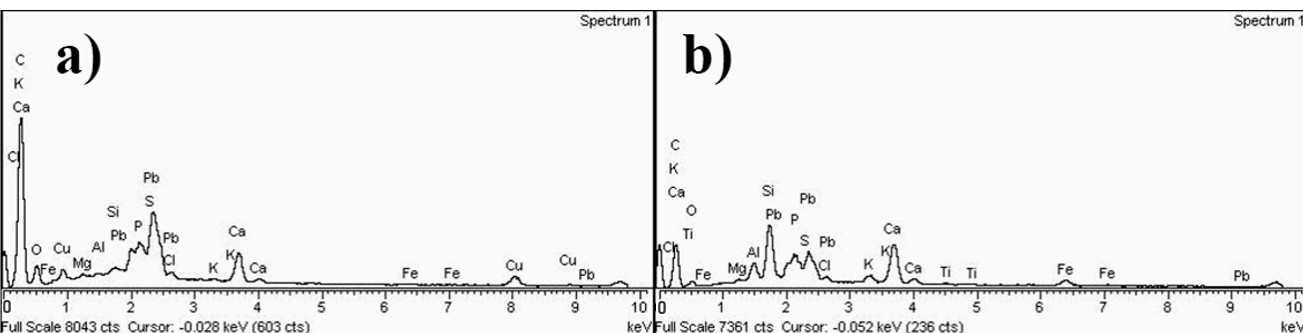

**Figure 4.** (**a**) SEM-EDX spectrum corresponds to the top blue layer in sample S1; (**b**) SEM-EDX spectrum corresponds to the blue layer in sample S3.

The sample S2 (Figure 1b) was taken from the outer edge of the icon. The cross-section of the sample S2 in OM in visible light (Figure 3d) shows a thick white preparation layer A, followed by thin orange layer B (preparation layer for the silver leaf techniques), very thin silver leaf (layer C), and a uniform dark-blue-coloured layer D on top. SEM image of layer D (Figure 3f) shows smooth edge microstructure [7] and thus appoints the employment of Prussian blue.

The intense red areas in the IRFC image of the icon GMSU 1451—*St. John the Baptist with Scenes from His Life* (Figure 1f) suggest the presence of either organic indigo or natural ultramarine. The absence of concurrent existence of chemical elements (Na, Al, Si, and S) in the SEM-EDX spectrum (Figure 2i) characteristic for natural ultramarine confirmed the use of organic indigo pigment. Moreover, the dark-blue lines delineate the contours of the draperies and architectural details which are clearly visible on Figure 1f as dark-blue-blackish lines suggesting the use of blue pigment other than indigo.

The absence/low intensity of any characteristic peaks in the XRF spectra (Figure 2g) confirms the use of organic pigments, while the rFTIR lines doublet around 1460 and 1480 $cm^{-1}$ originating from C-H bending and C-C stretching combination, and lines around 1580 ascribed to CC stretching vibrations in a six-membered ring of the indigo molecule and 1630 $cm^{-1}$ caused by the C=O stretching (Figure 2h) reveal the use of indigo [23].

Sample S3 (Figure 1c) was taken from the blue spot at the lower left corner area of the icon, next to the existing damaged part. The OM of the cross-section of sample S3 under UV light is presented in Figure 3g for better visualization of fine layers of different colours. The bottom white layer A corresponds to the preparation layer. On top of it, six different layers (B–G) can be discerned. Layers B, C, D, and E contain fine dark-blue particles, indicative of indigo. This is confirmed by EDX spectra, containing an intense peak of carbon (Figure 2i, corresponding to the layer E in the OM picture) and by the appearance of a sponge-like structure in the corresponding layer on the SEM figure (Figure 3i). Layer F, which is milky white in UV light, is an organic isolating layer, followed by the top pigmented layer G. Both EDXRF and EDX spectra of the layer G (that corresponds to spectrum 1 in Figure 3h) display a low, intense peak of iron (Figure 2g and Figure 4b, respectively), indicating the presence of Prussian blue, therefore, the blue areas were painted with indigo, while the contours were painted in Prussian blue. The finding has been confirmed by the rFTIR spectra, where clearly visible lines around 2090 $cm^{-1}$ (Figure 2h) have been noticed. Egg tempera has been detected as a binder used in the Icon GMSU 1451 as well.

On the five icons (GMSU 6215, 6334, 6432, 6472, and 1586), copper-based blue pigment azurite was identified. Indigo was proven on two icons (GMSU 1451 and 6506), mostly used together with Prussian blue. Iron-based Prussian blue was identified on four icons (GMSU 1451, 6112, 6432, and 6506). All collected analytical data are presented in Table 2.

## 4. Conclusions

Three blue pigments were identified: azurite, indigo, and Prussian blue. Among the identified inorganic blue pigments, azurite prevails since it was identified on five of the eight selected icons. Although azurite was less present on the palette of Western European painters in the 18th century, azurite remained in common use for the whole first half of the 18th century in the practice of Serbian traditional icon painters. A detailed analysis such as this one enabled us to conclude that the identified azurite was of artificial origin. The results also confirm that the widespread use of indigo remained in the post-Byzantine period. Prussian blue, the new pigment of the 18th century, was quickly adopted by traditional icon painters, offering affordable enrichment of the palette. The chemical composition of Prussian blue might indicate the manufacturing process used in pigment production since a considerable amount of phosphorus has been detected. Expensive natural ultramarine blue was not found, neither in the set of icons analysed on this occasion nor in other similar studies. It is understandable, as the icons of the first part of the 18th century were the icons made for common people and not for wealthy clients.

**Supplementary Materials:** The following supporting information can be downloaded at: https://www.mdpi.com/article/10.3390/coatings13071200/s1, Figure S1. The comparative EDXRF spectra collected at the blue parts of the icon GMSU 6472. Zoom in the ~1–10 keV energy range and 2000 counts; Figure S2. The comparative EDXRF spectra collected at the blue parts of the icon GMSU 6112. Zoom in the ~1–10 keV energy range and 1200 counts; Figure S3. The comparative EDXRF spectra collected at the blue parts of the icon GMSU 1451. Zoom in the ~1–10 keV energy range and 800 counts.

**Author Contributions:** Conceptualization, M.G.-K.; methodology, M.G.-K. and O.K.; formal analysis, M.G.-K., V.A., S.R. and U.G.; investigation, O.K. and D.K.C.; resources, D.K.C.; writing—original draft preparation, M.G.-K., O.K. and D.K.C.; writing—review and editing, S.R.; funding acquisition, V.A., S.R. and M.G.-K. All authors have read and agreed to the published version of the manuscript.

**Funding:** This research was funded by the Ministry of Science, Technological Development, and Innovations, Republic of Serbia (Grant No. 451-03-47/2023-01/200017-Research Programme No. 1-Contract No. 110-10/2019-000, T0602303, and Grant No. 451-03-47/2023-01/200125). Olivera Klisurić acknowledges the partial financial support of the APV Provincial Secretariat for Higher Education and Scientific Research (Project no. 142-451-3154/2022-01/2).

**Institutional Review Board Statement:** Not applicable.

**Informed Consent Statement:** Not applicable.

**Data Availability Statement:** Not applicable.

**Acknowledgments:** The authors express gratitude to the Gallery of Matica Srpska which supported the research and gave consent for the analysis. The authors are also grateful to professor Miloš Bokorov from the Electron Microscope Laboratory of the University of Novi Sad, and to Milica Marić Stojanović from the Scientific Laboratory of the National Museum in Belgrade for their generous help during the microscopic image acquisition.

**Conflicts of Interest:** The authors declare no conflict of interest.

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
