# Peer review of "Multianalytical Study of the Blue Pigments Usage in Serbian Iconography at the Beginning of the 18th-Century"

_coatings, doi:10.3390/coatings13071200_

Round 1

Reviewer 1 Report

Research is focused on analytical study of the paint component (blue pigments), using non-destructive modern analytical techniques. The work does succeed in characterizing icon painting  from specific period which could be important information for researchers or conservators. Furthermore the work may serve as a model for other researchers in the field of cultural heritage, who seek to characterize specific materials used in the specific period. This is the strength of the manuscript.

Authors have provided adequate literature review, up to date information and they incorporated findings from the literature appropriate in the discussion. The findings are presented clear in well-organized manner. Tables and figures are well integrated in the text. Discussion is well organized, practical significance of the research is emphasized. Research results are original and significant. Article is well structured, well written, clearly and concisely and is easily understood.

I recommend acceptance of the work.

Author Response

The authors are grateful for the reading our manuscript and the comments, which have encouraged the presentation of the results of our research.

Kind regards,

Maja Gajić-Kvaščev, PhD

Associate Research Professor

Vinča Institute of Nuclear Sciences

Reviewer 2 Report

The work is original. Its publication is recommended after the following points have been considered/improved:

Abstract

Lines 17-18: the expression “historic migration of the nation” is not understandable by a non-specialist in the history of Serbia. To clarify or delete.

Line 29: Replace “ultramarine blue” with lapis lazuli, the term ultramarine is generally used for modern synthetic zeolite imitating lapis lazuli.

Text and Figures

Paragraph 2.3, line 120: the resolution varying with the energy it would be useful to also give the resolution for a heavier element. It should be clearly explained whether the XRF is done on the (upper) surface of the coloured areas or on the polished sections. Indeed, as recalled in recent studies because forgotten by many authors, the depth analyzed in XRF varies with the energy of the photons from a few µm for light elements to several mm for heavy ones (see e.g. doi:10.3390/heritage. 3040072), that have important effects on the spectrum.

Figure 2 is illegible, either the writings or the spectra. It is recommended to cut it in two part, on one side the elementary analysis spectra (magnified), on the other the IR spectra and to use larger fonts. Add a table listing the characteristic wavenumbers and their attribution as well as the references used for the attribution. Note that in IR reflection it is the point of inflection and not the peak which is the characteristic position of the mode of vibration but unfortunately the measurements in reflection on a porous surface lead to the combination of spectra of reflection stricto sensu ( that can be transformed in absorption spectrum with KK law) and absorption (improperly called diffuse reflectance). The text needs to be rephrased.

Figure 3: the “spectrum X” indications are difficult to read; improved contrast and colour balance would make this figure more useful.

Lines 230 and following. Only the Raman analysis, because of its exceptional sensitivity to the detection of the chromophore Sn ions, makes it possible to know whether lapis or ultramarine is used. The presence of potassium for one of the spectra questions the presence of smalt. For an effective reading, the presentation of the spectra in Figure 2 must be improved by presenting the domain amplified up to 1000 or 2000 counts so that the information will be not obscured by the intensity of the L peaks of the lead. This would make it possible to compare the signal around 7 keV where the Fe Kbeta and Co Kalpha contributions are superimposed (the instrumental resolution is not good enough to separate them). Visual comparison of Fe Kalpha and FeKbeta-CoKalpha peak intensities would show if smalt was used as one of the 2d spectra suggests [doi:10.3390/heritage5030091]. Other elements are present (Zn or Ni?). The text needs to be rephrased and minor peaks better discussed.

In Table 2 only information proving the alleged pigment should be given. The identification of phosphorus must be discussed.

Reformulate the conclusion and the abstract as consequences of the remarks.

OK

Author Response

The authors are grateful for the comments and suggestions, which have helped improve the presentation of the results of our research.

We submitted our answers in the separate file, which we attached. 

Kind regards,

Maja Gajić-Kvaščev, PhD

Associate Research Professor

Vinča Institute of Nuclear Sciences

Reviewer 3 Report

The presented paper deals with very interesting but sensitive topic as it is aimed to determine characteristics of art. The performed analysis is well structured but it is hard t ofollow as it mentiones figures 2, 3, 4 even before Figure 1 is presented. In the abstract authors indicate that the choice of the pigment can give historical information, but do not indicate correlation to the results presented. In addition, I would suggest:

- more sources in the introduction (for example for the statement in paragraph 2, l38-43)),
- l75 - the abreviation should be SEM-EDS;
- table 2 - give more sources for FTIR spectral data

The English should be omproved in some grammar and style.

Author Response

(The authors gave the same response as above.)

Reviewer 4 Report

The work focuses on the blue pigments on 8 icons from the first half of the 18th century. Several portable, non-destructive analytical techniques such as XRF and FTIR, followed by optical microscopy and SEM-EDX analysis were used in the investigation. The analysis showed that icon painters of the early 18th century used pigments such as azurite, organic indigo and Prussian blue, instead of ultramarine blue, not being confirmed in any sample of pictorial material.

I am very pleased with the value of the paper, which I find attractive.

The summary is well worded.

The literature has been well selected (using 23 bibliographic references) and the critical analysis of the references is well developed.

The working methods and instrumental techniques are up-to-date and allow the elucidation of the nature of the component materials of the pictorial layer of the eight Serbian Orthodox icons of the century. xviii

The figures (photos), graphs and tables are very well done, and the interpretation of the experimental results corresponds to the level agreed by the journal.

As for the conclusions, they summarize the obtained results very well.

As stated, the bibliographic references (23) are rigorously selected from the current specialized literature.

The authors studied the blue pigments in the color layer of eight old icons to elucidate their chemical nature, without finding ultramarine blue, often used in fresco and painting of old icons on wood.

The work is interesting, but it would have been good if a column with the narrowed images of the 8 old icons was inserted in table 1.

Author Response

The authors are grateful for the reading our manuscript and the comments, which have encouraged the presentation of the results of our research.

Table 1 has been corrected according to the suggestions.

Kind regards,

Maja Gajić-Kvaščev, PhD

Associate Research Professor

Vinča Institute of Nuclear Sciences